# Emerging Roles for the *INK4a/ARF* (*CDKN2A*) Locus in Adipose Tissue: Implications for Obesity and Type 2 Diabetes

**DOI:** 10.3390/biom10091350

**Published:** 2020-09-22

**Authors:** Yasmina Kahoul, Frédérik Oger, Jessica Montaigne, Philippe Froguel, Christophe Breton, Jean-Sébastien Annicotte

**Affiliations:** U1283-UMR8199-EGID, Univ. Lille, INSERM, CNRS, CHU Lille, Institut Pasteur de Lille, F-59000 Lille, France; yasmina.kahoul.etu@univ-lille.fr (Y.K.); frederik.oger2@univ-lille.fr (F.O.); jessica.montaigne@univ-lille.fr (J.M.); p.froguel@imperial.ac.uk (P.F.)

**Keywords:** adipogenesis, insulin sensitivity, inflammation, obesity, oxidative activity, senescence, type 2 diabetes

## Abstract

Besides its role as a cell cycle and proliferation regulator, the *INK4a/ARF* (*CDKN2A*) locus and its associated pathways are thought to play additional functions in the control of energy homeostasis. Genome-wide association studies in humans and rodents have revealed that single nucleotide polymorphisms in this locus are risk factors for obesity and related metabolic diseases including cardiovascular complications and type-2 diabetes (T2D). Recent studies showed that both p16^INK4a^-CDK4-E2F1/pRB and p19^ARF^-P53 (p14^ARF^ in humans) related pathways regulate adipose tissue (AT) physiology and adipocyte functions such as lipid storage, inflammation, oxidative activity, and cellular plasticity (browning). Targeting these metabolic pathways in AT emerged as a new putative therapy to alleviate the effects of obesity and prevent T2D. This review aims to provide an overview of the literature linking the *INK4a/ARF* locus with AT functions, focusing on its mechanisms of action in the regulation of energy homeostasis.

## 1. Introduction

The global epidemic of obesity and associated metabolic diseases have increased substantially over the past decades. Obesity is defined as an increased body weight characterized by an excessive accumulation of fat in adipose tissue (AT) mainly because of an imbalance between energy intake and energy expenditure [1]. Indeed, although genetics may explain parts of the variation in body mass, environmental factors such as overnutrition and ultra-processed foods, sedentary lifestyle, xenobiotics, as well as chemical exposure are the major contributors to the rapidly increasing prevalence [2]. Energy balance regulation depends on extremely complex processes that integrate multiple interacting hypothalamic neural pathways and metabolic signaling mechanisms (i.e., hormones, nutrients, sensory, and nerve inputs) typically referred to as the homeostatic regulation but also engage most other parts of the brain referred to as non-homeostatic regulation. It also depends on key cellular energy sensors (such as AMP kinase) which act as an integrator of regulatory signals monitoring systemic and cellular energy balance [3]. The long-term excessive AT mass, progressively leads to the appearance of a marked inflammatory profile and to its secondary dysfunction (i.e., impaired expandability and plasticity) [4]. The inability of further AT expansion accelerates fat spillover from AT to skeletal muscle and liver, resulting in ectopic fat deposition and insulin resistance in these tissues, which play vital roles in systemic insulin resistance and T2D [5,6]. These modifications that strongly contribute to T2D and obesity comorbidity development [7,8], makes AT a putative new therapeutic target.

Two anatomically and functionally distinct types of AT exist in mammals: the white (WAT) and brown (BAT) AT. The AT cellular content is heterogeneous and includes adipocytes, preadipocytes, fibroblasts, immune cells, and endothelial cells [9]. White adipocytes are characterized by the presence of a wide single (unilocular) lipid droplet. They store energy excess as triglycerides (TG) and release free fatty acids as energy substrate during period of negative energy balance. WAT exists in multiple locations in the body with two major subtypes: visceral and subcutaneous [10]. Adipose depots are composed of adipocytes that derive from distinct precursor populations and contain progenitors through distinct lineages [11]. In normal growth throughout life and during obesity, the renewal/expansion of AT relies on both hyperplasia (increase in number via adipogenesis) and hypertrophy (increase in size via TG storage) which contribute to maintaining AT in a healthy state [10]. Unlike visceral WAT (vWAT), the metabolic adaptability of subcutaneous WAT (sWAT) to changes in its environment, a process called plasticity, has been associated with increased insulin sensitivity and decreased rates of T2D [12]. 

BAT specializes in energy expenditure and production of heat, mainly via active fat oxidation. In contrast to white adipocytes, brown adipocytes contain numerous (multilocular) smaller lipid droplets and a much higher number of mitochondria. Brown adipocytes are characterized by a high expression of the thermogenic uncoupling protein 1 (UCP1), a BAT-specific marker. BAT differs from WAT by its cellular origin, sharing a common progenitor with skeletal muscles, both expressing the myogenic regulatory factor Myf5 [13]. Recent studies have revealed a new distinct type of thermogenic adipocyte—named beige cells (also known as brite cells for brown in white)—intermingled within WAT [14,15,16]. Beige adipocytes might arise via de novo differentiation from undifferentiated adipocyte-derived stem cells or via a transdifferentiation of existing mature white adipocytes. Beige adipocytes also express UCP1 and mainly develop under cold exposure or in response to noradrenergic stimulation through the β3 adrenergic stimulation. This adaptive AT plasticity is called WAT beiging or browning [17]. In adult humans, the abundance of brown/beige fat is reduced with obesity and the challenge is to prevent its loss with aging or to reactivate existing depots or both. Beige fat has the largest potential as a therapeutic target in the prevention of obesity and T2D because it can be present in many white depots as clusters of pre-adipocytes that then can be recruited [11,12].

Genome-wide association studies (GWAS) have established the cyclin-dependent kinase inhibitor *(CDKN)2a* locus, also called *INK4a/ARF*, as a hotspot influencing genetic risk for cardiovascular and metabolic diseases including T2D. Single nucleotide polymorphisms (SNPs) and loss-of-function mutations in *CDKN2A* locus affect glucose values, insulin sensitivity, and T2D risk [18,19,20,21]. Epigenetic modifications are also likely to play an important role in regulating this locus. Epigenetics is defined as heritable changes in gene expression that do not involve changes to the underlying DNA sequence. Epigenetics relate to chromatin modifications, including DNA methylation and histone modification [22]. In particular, decreased DNA methylation within the promoter of ANRIL (Antisens Noncoding RNA in the INK4 Locus), a 3.8 kb non-coding RNA transcribed from the genomic INK4b-ARF-INK4a locus, is associated with a higher risk for a child to develop obesity during adulthood [23]. Consistently, deletion of a region of mouse chromosome 4, orthologous to the human 9p21 metabolic disease risk interval, results in increased body weight, linking the region with obesity and metabolic syndrome [24]. 

The human *CDKN2A* locus spans around 35 kilobases on chromosome 9p21 (Figure 1). It encodes for two proteins, p16INK4a, the principal member of the INK4 family of cyclin-dependent kinase inhibitors (CDKI) and the p53 regulatory protein p14ARF. While p14ARF is transcribed from exon 1b and exon 2, p16INK4a is transcribed from exon 1a localized 20 kb downstream of 1b and exons 2 and 3. p16INK4a binds to CDK4/6, inhibiting cyclin D-CDK4/6 complex formation and CDK4/6-mediated phosphorylation of Rb family members (pRB). Expression of p16INK4a maintains pRB in a hypophosphorylated state, which promotes binding to the transcription factor E2F1 and blocks the progression of the cell-division cycle. p14ARF (p19ARF in mice), which is an alternate reading frame protein product of the *CDKN2A* locus, mainly exerts its anti-proliferative activity via the inhibition of the mouse double minute 2 homolog (MDM2), an ubiquitin-ligase that hampers the activity of the transcription factor p53, acting as a tumor suppressor [18,19] (Figure 1). 

In line with its canonical role in cell-cycle progression and differentiation, several studies showed that the *INK4a/ARF* locus promotes AT development [25,26,27,28]. Two main adipogenesis phases are described: the first phase, called commitment, results in the conversion of the progenitor proliferating cells into preadipocytes [29]. This phase is then followed by terminal differentiation, during which specified preadipocytes take on the characteristics of the mature adipocyte [30]. Adipogenesis involves a complex orchestrated gene expression program under the control of the transcription factors CCAAT-enhancer-binding proteins (CEBPs) and peroxisome proliferator-activated receptor (PPAR)γ that induces mature adipocyte formation [31,32,33].

First, p16INK4a knock-down in 3T3-L1 increases adipogenesis [34]. p16INK4a-deficient mouse developed more epicardial AT in response to the adipogenic PPARγ agonist rosiglitazone, providing a potential mechanistic link between the genetic association of the *INK4a/ARF* locus and cardiovascular disease risk [34,35]. CDK4 also directly phosphorylates PPARγ, thus promoting terminal differentiation [36]. Second, E2F1 promotes preadipocyte differentiation by activating inhibitor of β-catenin and TCF4 (ICAT), therefore repressing Wnt/β-catenin activity in 3T3-L1 cells [37]. E2F1 complex-dependent transcription repression by physical association of C/EBPα inhibits the proliferation of progenitors and induce adipocyte terminal differentiation [38,39]. It activates the PPARγ promoter triggering PPARγ expression during adipogenesis [38]. Accordingly, E2F1-deficient primary embryonic fibroblasts (MEF) have a reduced capacity to differentiate into adipocytes and E2F1-deficient mice are resistant to diet-induced obesity (DIO) [38]. Third, pRB deletion in MEFs results in enhanced number of committed preadipocytes. During the early stages of adipogenesis, pRB interacts with PPARγ to inhibit adipocyte differentiation [40]. Thereafter, pRB activates adipogenesis for terminal differentiation through association with C/EBPα [41,42]. pRB deletion in 3T3-L1 cells fails to undergo terminal differentiation [42] whereas overexpression of pRB promotes terminal adipocyte differentiation [43]. Four, p53 inhibits adipogenic differentiation [44]. p53-deficient MEFs undergo spontaneous commitment to the adipogenic program and ectopic re-expression of p53 inhibits their spontaneous differentiation [45,46]. Nutlin-3a-mediated p53 accumulation (by destabilization of the p53/MDM2 complex) in MEFs leads to decreased PPARγ [47] whereas the knockdown of p53 in 3T3-L1 results in enhanced PPARγ and differentiation [46,47,48]. The p53-deficient mice high sensibility to DIO may rely on the regulatory role of p53 as an inhibitor in the process of white versus brown fat accumulation [46].

Alternatively, *INK4a/ARF* locus is thought to regulate additional functions in adipocyte. In this review, we will present data supporting the role of the *INK4a/ARF* locus as a key regulatory hub to maintain AT in a healthy state. As indicated in Figure 2, this locus regulates the balance between adipogenesis and senescence [49,50,51,52,53] and plays a role in adipocyte insulin sensitivity and lipid storage [34,36,54,55], inflammation [56,57,58] as well as oxidative activity and browning [59,60,61]. Targeting these metabolic pathways have recently emerged as a new putative therapy to alleviate the effects of obesity and prevent insulin resistance and T2D [62].

## 2. The *INK4a/ARF* Locus: A Balance between Adipogenesis and Senescence

Emerging evidence from human and animal studies suggests that the *INK4a/ARF* locus controls the balance between adipocyte differentiation and senescence [49,50,51,52,53] (Figure 2). Like aging, obesity can be seen as an accelerated form of AT senescence [63]. In both cases, cell senescence characterized by elevated p16INK4a/p53 expression appears to be a key mechanism for reduced regenerative potential of adipose progenitor cells and impaired adipocyte replacement. However, unlike aging, MDM2 expression is paradoxically increased in obese AT, indicating that the pathways leading to p53 activation in aged and obese AT may be distinct [63]. In addition to defective adipogenesis, in both obesity and aging, the immune profile dramatically alters to inflammatory status and associates with number of senescent cells, aberrant adipocytokines production, leading to local (AT) and systemic insulin resistance and T2D [49,50,51,52,53,64].

Indeed, the differentiation of new adipocytes is associated with marked improvement of insulin sensitivity, mostly through increased fatty acid buffering capacity, modification of adipokine repertoire, and possibly through increases in the amount of cellular membranes that act as a buffering system for cholesterol esters and other lipid signaling molecules [65]. Although metabolic insults (such as high glucose and saturated fatty acid) directly up-regulate p16INK4a/p53 expression, yet the underlying mechanisms are still unclear. In addition, it remains elusive whether the cell cycle changes are causative or resultant of AT dysfunction [63]. In short, the defective adipogenesis is mainly due to poor-differentiating ability of senescent progenitor cells and senescent microenvironment within AT in obesity and aging. The mechanistic basis for aging or obesity-associated adipose stem cell decline linked to elevated p16INK4a/p53 expression is not completely understood. However, increasing evidence suggests that epigenetic dysregulation is an important mechanistic driver of stem cell fate during these processes [66]. Thus, it is tempting to speculate that epigenetic mechanisms regulating differentiation transcriptional program may account for the adipose stem cell fate decision between senescence and differentiation in obese individuals. These findings are in line with the AT expandability theory which proposes that the inability of further AT expansion in obese individuals is a key determinant of dysfunctional AT, inflammation, and insulin resistance [65]. During obesity, the enlargement of the vascular network is not sufficient to supply enough oxygen to hypertrophic adipocytes in obese individuals resulting in local hypoxia. This hypoxia could be a key trigger of AT dysfunction (i.e., fibrosis) and inflammation by induction of gene expression in adipocytes and macrophages [67].

A recent study by Gustafson et al. shed light on the involvement of this locus in reduced adipogenesis in AT of obese patients [49]. They reported that individuals with hypertrophic obesity and insulin resistance displayed a dysfunctional AT with inappropriate expansion of the adipose cells in sWAT. Interestingly, progenitors undergoing poor differentiation have elevated p16INK4a/p53 expression which is a characteristic of senescent cells. Diminishing senescence and P53 activity in dysfunctional AT may constitute an alternative method to prevent insulin resistance and T2D. Indeed, reducing senescence either genetically or with senolytic agents in obese mouse models with T2D restores adipogenesis in sWAT and alleviates metabolic and AT dysfunction (i.e. reduction of inflammation and enhanced insulin sensitivity) [50,51]. Increased p53 has also been reported in AT from individuals with T2D and overexpression of p53 in AT in rodent models triggers insulin resistance and inflammation [52,53,64]. Like senescence, inhibition of p53 activity in AT protects obese mice from insulin resistance and reduce AT inflammation [53].

## 3. The *INK4a/ARF* Locus: A Role in Adipocyte Insulin Sensitivity and Lipid Storage

Several studies showed that the *INK4a/ARF* promotes TG accumulation and adipocyte hypertrophy via the insulin-signaling pathway (Figure 2). In particular, the cyclin-dependent kinase 4 (CDK4) which mediates the phosphorylation of pRB resulting in the dissociation of E2F1 and activation of the cell-cycle progression from G1 to S phase (Figure 1) is a key molecule involved in adipocyte insulin sensitivity. CDK4 acts independently of E2F1 in a cell-cycle-independent manner to regulate lipid storage in mature adipocyte [34,36,54,55].

CDK4 is expressed in fully differentiated adipocytes in both human and mouse AT under physiological conditions [36]. CDK4 activity in AT is positively correlated with fat mass, lipogenesis, and insulin sensitivity. Pharmacological inhibition of CDK4 in fully differentiated 3T3-L1 adipocytes decreased insulin sensitivity, reduced glucose transport and lipogenesis resulting in smaller lipid droplets [34,36,54,55]. In line with these observations, CDK4-deficient mice have less fat mass and smaller adipocytes [68]. They also exhibit impaired insulin signaling and lipogenesis in addition to insulin deficiency, the latter being due to reduction of the pancreatic β-cell number [69]. CDK4 re-expression in β-cells does not rescue body weight reduction and still displays hypotrophic adipocyte suggesting cell-autonomous contribution for CDK4 in AT [70]. Consistently, mice expressing a hyperactive CDK4 mutant have increased body weight, fat mass, and larger adipocytes [68]. As strong evidence of its cell-autonomous action, CDK4 was shown to activate the insulin-signaling pathway through phosphorylation of insulin receptor substrate 2 (IRS2) at Ser388 upon insulin stimulation, thus maintaining insulin action in adipocytes ([55], Figure 3). Interestingly, IRS2-Ser388 phosphorylation in human vWAT is positively correlated with BMI and negatively correlates with blood glucose levels in patients [55].

In addition, elevated E2F1 contents in adipocytes of vWAT from obese patients are positively correlated with insulin resistance. Besides its role in cell-cycle progression, E2F1 also activates promoter activity of several autophagy genes [71,72]. Thus, E2F1 adipocyte levels might play a role in linking obesity with AT insulin sensitivity by upregulating autophagy gene expression and by sensitizing cells to inflammation [71,72]. In agreement with this notion, MEFs-derived adipocytes from E2F1-deficient mice exhibited increased insulin sensitivity with lower inflammation and autophagy [72]. Finally, treatment of 3T3-L1 and human adipocytes with p53-inducers doxorubicin (a DNA damage-inducing drug) and nutlin-3a reduces insulin-stimulated glucose uptake by lowering GLUT4 glucose transporter translocation. This adipocyte cell-autonomous effect occurs independently of inflammation [64]. Although E2F1 and p53 are thought to interact with the insulin-signaling pathway to modulate AT insulin sensitivity, yet the underlying mechanisms are still unclear.

## 4. The *INK4a/ARF* Locus: A Role in Adipose Tissue Inflammation 

Obesity provides a plethora of intrinsic and extrinsic signals capable of triggering an inflammatory response in AT resulting in low-grade chronic AT inflammation. These mechanisms are commonly considered the link between chronic caloric excess and AT meta-inflammation. Some of these mechanisms include dysregulation of fatty acid homeostasis, increased adipocyte size and death, local hypoxia, mitochondrial dysfunction, increased reactive oxygen species production, endoplasmic reticulum, and mechanical stress leading to extracellular matrix remodeling and fibrosis [5,6]. These triggers converge on the activation of the c-Jun N-terminal kinase (JNK) and nuclear factor-kappa B (NF-kB) pathways, commonly considered signaling hubs. The activation of these pathways increases the production of pro-inflammatory cytokines and promotes the recruitment of a variety of immune cells, including macrophages, T cells, B cells, neutrophils, and others. These immune cells produce a large number of type 1 inflammatory molecules and also interact directly with each other to induce a type 1 inflammatory environment in AT resulting in local (AT) and systemic insulin resistance and T2D [5,6]. Therefore, inflamed AT is characterized by the combination of an increase in total macrophages and an increased ratio of «classically-activated» proinflammatory M1 phenotype versus «alternatively-activated» anti-inflammatory M2 phenotype macrophages [4,73] that is typical of lean individuals [74]. Emerging evidence suggests that the *INK4a/ARF* locus is involved in macrophage phenotype and thus obesity-related inflammation [57,75] (Figure 2).

Although p16INK4a deficiency does not affect macrophage proliferation or maturation, its anti-inflammatory effects contribute to the polarization of macrophage by inducing a shift of AT-associated M1 macrophages toward the M2 state. Thus, bone-marrow-derived macrophages (BMDMs) isolated from p16NK4a-deficient mice exhibit a phenotype resembling «alternatively-activated» anti-inflammatory M2 with low expression levels and secretion of inflammatory cytokines [56,57]. In line with these observations, AT macrophages from overweight patients express higher levels of p16INK4a. The silencing of p16NK4a in macrophages from these patients mediated by siRNA increases the expression of M2 marker genes to give a phenotype resembling that of macrophages from healthy donors [57].

Several studies have also highlighted a role for p53/MDM2 complex as a physiological “brake” on M2 polarization. By using BMDMs, p53 activity increased when macrophages were polarized to the M2 state facilitating expression of M2 genes. However, forcing further activation of p53 using nutlin-3a downregulated M2 gene expression [58]. In agreement with these findings, M2-polarized macrophages from p53-deficient mice showed increased expression of M2 genes [58]. In addition, p53 induces upregulation of semaphorin, which acts as a chemoattractant for macrophages in vWAT [76]. Although upregulation of both p53 activity and inflammation is commonly observed in WAT of obese rodents and humans, the causal relationship remains unclear [44]. It is tempting to speculate that the age- and obesity-related increase in p16INK4a and P53 levels in AT [49,52,53,64], not only facilitates the recruitment of macrophages in AT, but also hampers macrophage M2 polarization and contributes to T2D risk. 

## 5. The *INK4a/ARF* Locus: A Role in Adipose Tissue Oxidative Activity and Browning 

Originally, data highlighted the importance of silencing the *INK4a/ARF* locus to allow for proliferation and reprogramming [77]. This locus is considered as a general suppressor of differentiation in embryonic stem cells and of reprogramming in induced pluripotent stem cells (iPSCs), controlling cell fate determination and plasticity [78]. 

Recently, the *INK4a/ARF* locus has been suggested to play a major role in the molecular switch of white-to-beige adipocyte conversion and thus is a key determinant of brown adipocyte fate (Figure 2). In support of this notion, we showed that *CDKN2A*-deficient mice were protected against DIO exhibiting sWAT browning and increased thermogenesis and insulin sensitivity [60]. Differentiated SVF cells from sWAT displayed elevated oxidative activity with upregulation of brown fat-specific gene and increase in PKA activity and adrenergic receptor signaling pathways. This suggests a cell-autonomous requirement of *CDKN2A* to the browning process. Consistent with these findings, silencing *CDKN2A* expression during human iPSCs adipogenic differentiation promotes UCP1 expression and browning markers [60]. We also observed that *CDKN2A* expression was increased in adipocytes from obese patients. Whether p16INK4a and p19ARF pathways modulate AT plasticity through transdifferentiation of fully differentiated adipocytes remains elusive. However, inducible ablation or pharmacological inhibition of p53 in mature adipocytes of mice restore cold-induced beiging and increase energy expenditure and insulin sensitivity [79].

These data raise the hypothesis of potential effects of *CDKN2A* ablation in adipocyte progenitor cells. In line with these findings, blocking these pathways, either genetically (deletion of the *INK4a/ARF* locus) or pharmacologically, in aged mouse or human beige dysfunctional progenitor cells, reverses the senescence-like process and restores their potential to form cold-induced beige adipocytes leading to increased insulin sensitivity [80]. In this respect, *CDKN2A*-deficient mice restricted to adipocyte precursors should assess the contribution of *CDKN2A* in modulating adipose stem cells beiging. 

Somatic mutations in *CDKN2A* or dysregulation of its functional activity are frequently detected in various types of human cancer. Studies in rodents and humans reported that cancer-associated cachexia triggers sWAT browning and may contribute to increased oxidative activity, energy expenditure and weight loss [81,82]. Therefore, we cannot rule out the fact that the inactivation of *CDKN2A* in cancer cells may indirectly account for AT browning (at least partly via β3-adrenergic activation) participating in the development of cachexia.

The E2F1/pRB repressor complex was proposed to act as a molecular switch of cellular utilization from glycolytic to oxidative metabolisms, hence in the adaptation to energy demand. On the one hand, chromatin immunoprecipitation has demonstrated binding of E2F1 and pRB to gene promoters involved in oxidative metabolism. On the other hand, an E2F1 and pRB complex is required for the repression of the transcription of these genes. For instance, pRB was observed to bind the promoter of PGC-1α, the mitochondrial biogenesis master regulator, to repress transcription [59,83]. As reported for *CDKN2A*-deficient mice [60], E2f1- and adipocyte-specific pRB-deficient animals [59,61] displayed resistance to DIO associated with a marked mitochondrial oxidative activity with enhanced fatty acid oxidation-related genes, thermogenesis and insulin sensitivity. Browning induction of sWAT and elevated energy expenditure may account for the resistance to DIO with improved insulin sensitivity in both *CDKN2A*- and pRB-deficient mice [60,61]. Consistent with these findings, pRB-deficient MEFs exhibited BAT characteristics, with activation of BAT-specific genes and an increase in mitochondria activity [61]. 

Discordant findings have been reported in the literature regarding the implication of P53 in BAT differentiation and activity [44]. On the one hand, Molchadsky et al. showed that, although p53 suppresses white adipogenic differentiation, it seems to be required for brown adipogenic differentiation. Indeed, BAT from adult obesity-prone p53-deficient mice displayed abnormal morphology and reduction in expression levels of key genes for brown adipocyte function suggesting that p53 might be involved, not only in brown fat differentiation and development, but also in maintaining the full integrity of a brown fat phenotype [46]. On the other hand, p53 has emerged as a negative regulator of brown adipogenic differentiation from progenitor cells. Indeed, Hallenborg et al. reported that brown differentiation efficiency of MEFs and primary adipocytes, both deriving from p53-deficient mice, was enhanced, evidenced by higher expression of UCP1 mRNA levels [45]. p53 impairs oxidative metabolism through inhibition of PGC1α activity [45,84]. Accordingly, mice bearing a global invalidation or having a reduced activity of p53 were resistant to DIO with browning of sWAT [45] or increased oxidative activity [85], respectively, resulting in increased thermogenesis and insulin sensitivity. However, in these studies, no differences in BAT morphology or activity were observed [45,85]. Overall, although P53 seems to be required in sustaining a proper brown adipocyte phenotype, the difference of p53-deficient mice phenotype regarding DIO sensitivity and BAT phenotype [45,46] may rely on differences in mouse strains and housing temperatures.

## 6. The *INK4a/ARF* Locus: An Emerging Key Actor in Metabolic Functions

In addition to AT, the *INK4A/ARF* (*CDKN2A*) locus is thought to play a variety of roles in metabolic functions under normal and pathological conditions (Figure 4). Several *CDKN2A* loss- or gain-of-function studies showed that this locus is involved in glucose homeostasis [86,87], β-cell functions (insulin secretion) and mass (proliferation) [88,89,90], gluconeogenesis [91], atherosclerosis [92,93,94,95], and hepatic steatosis [96]. A few studies also linked the *INK4A/ARF* locus to the regulation of the circadian rhythm via the modulation of RAS activity [97], neurogenesis [98], neuronal transdifferentiation [99], and axonal regeneration [100,101]. As reported in AT [60], the PKA activity appears to be a key target of this locus in liver and pancreas. p16INK4a-deficient mice enhanced fasting-induced hepatic glucose production through the activation of PKA-CREB-PGC1α signaling pathway by the phosphorylation of its regulatory subunits (PKAR2) independent of changes in intracellular cAMP levels [91]. *CDKN2A* knockdown in β-cell line results in an increase in insulin secretion accompanied by elevated PKA activity [87]. Premature atherosclerosis in T2D associates with altered immune cell homeostasis (i.e., T-cell subtype-associated proinflammatory state), diminished p16INK4a and ANRIL expression, and increased CDK4 levels. Treatment with drugs resulting in CDK4 inhibition via p16INK4a mimetic drugs are currently considered to delay atherosclerosis [102,103]. Ogrodnik et al. demonstrated that elimination of hepatic senescent cells via targeting the p16INK4A senescence regulator may be a novel therapeutic strategy to reduce hepatic steatosis [96].

ANRIL, also known as cyclin-dependent kinase inhibitor 2b-antisens RNA 1 (CDKN2b-AS1), also emerges as an actor in several metabolic diseases. CDKN2b-AS1 is transcribed under the influence of different transcription factors (e.g., E2F1 [104], SOX2 [105]) depending on the cell type and pathological conditions. SNPs within CDKN2b-AS1 locus are also associated with risks for several cancers [106]. From a functional point of view, CDKN2b-AS1 is directly involved in the regulation of gene expression in both cis and trans through its interaction with polycomb repressive complexes 1 and 2 (PRC-1 and PRC-2) [107,108]. CDKN2b-AS1 also possesses miRNA "sponge" functions [109]. Lillycrop et al. proposed that ERα could directly regulate the expression of CDKN2b-AS1 [23] suggesting that the well-known effects of ERα on glucose tolerance and insulin sensitivity in AT could be partly modulated by CDKN2b-AS1. Furthermore, given the important role of miRNAs in adipocyte functionality and modulation of their expression under certain pathological conditions [110], the fact that CDKN2b-AS1 plays a role with respect to its miRNA "sponge" function cannot be ruled out.

## 7. Discussion

The *INK4a/ARF* locus regulates adipocyte and AT functions, positioning this locus as an essential regulatory hub that participates in maintaining AT in a healthy state. As summarized in Figure 2, this locus is involved in the balance between adipogenesis and senescence and plays a role in adipocyte insulin sensitivity and lipid storage, inflammation, as well as oxidative activity and browning. Given that upregulation of the *INK4a/ARF* locus activity is commonly observed in dysregulated WAT of obese rodents and humans, downregulation and/or pharmacological inactivation of its activity may represent an alternative way to restore healthy AT. However, since p16INK4a, p14/p19ARF, and P53 are the anti-tumor pillars of the cell, their inactivation is challenging. On the other hand, the recent observations reporting that the *INK4a/ARF* locus plays a role in WAT browning raise the question of the pathways involved in the regulation of AT plasticity [17]. This alternative role is particularly interesting because increasing energy expenditure by promoting browning of WAT has recently emerged as a new putative therapy to alleviate the effects of obesity and prevent insulin resistance and T2D [11,12,111]. Indeed, the engraftment of BAT and beige/brown adipocytes, isolated from human adipocyte precursors and transplanted into mouse recipients, results in substantial weight loss, improvements in glucose tolerance, and insulin sensitivity, representing a clinically translatable model [112,113]. Therefore, it is challenging to identify the responsible cellular pathways and intermediates involved in browning independent of the cell cycle regulation. It might open up novel therapeutic options to protect against obesity consequences, insulin resistance and T2D. In particular, specific inhibitors, which have no effects on cell cycle progression but retain their ability to induce beiging, could be used to alleviate metabolic syndrome.

## Figures and Tables

**Figure 1 biomolecules-10-01350-f001:**
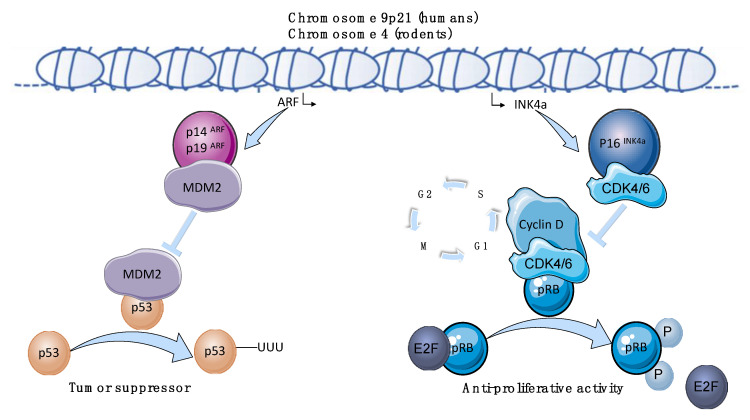
The *INK4a/ARF* locus. The *INK4a/ARF* locus is located on chromosome 9p21 in humans and chromosome 4 in rodents. It encodes for two proteins, p16INK4a the principal member of the INK4 family of cyclin-dependent kinase inhibitors (CDKI) and the p53 regulatory protein p14ARF (p19ARF in mice). Both are key regulators of the cell cycle machinery with an anti-proliferative and tumor suppressor role. p16INK4a binds to CDK4/6, inhibiting cyclin D-CDK4/6 complex formation and CDK4/6-mediated phosphorylation of Rb family members (pRB). Expression of p16INK4a maintains pRB in a hypophosphorylated state, which promotes binding to the transcription factors E2F and blocks the passage of the G1 to S phase. p14ARF (p19ARF in mice) mainly exerts its anti-proliferative activity via the inhibition of the mouse double minute 2 homolog (MDM2), an ubiquitin-ligase that hampers the activity of the transcription factor p53, acting as a tumor suppressor, blocking cells in G1 and G2 phase.

**Figure 2 biomolecules-10-01350-f002:**
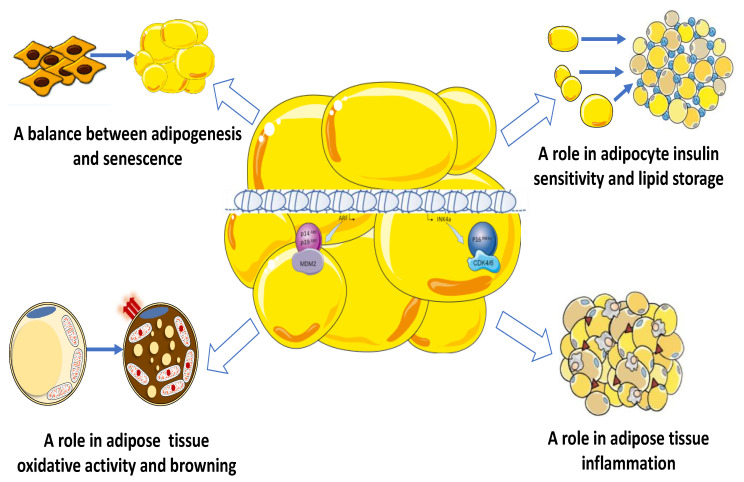
The *INK4a/ARF* locus as a key regulatory hub to maintain adipose tissue in a healthy state. The *INK4A/ARF* locus regulates the balance between adipogenesis and senescence and promotes lipid storage as triglycerides and adipocyte hypertrophy via the insulin-signaling pathway. It has been described as a molecular switch of white-to-beige adipocyte conversion and as a key determinant of brown adipocyte fate, being an alternative way to increase energy expenditure. It is also involved in the switch between macrophage phenotype and thus obesity-related inflammation.

**Figure 3 biomolecules-10-01350-f003:**
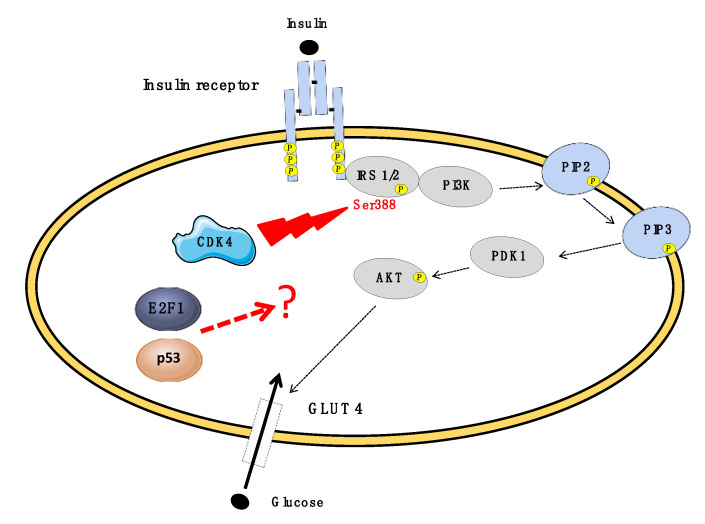
The *INK4a/ARF* locus modulates the insulin-signaling pathway in adipocyte. Insulin attaches to insulin receptor triggering intracellular autophosphorylation of their tyrosine residues, which constitutes an attachment for insulin receptor substrate (IRS) proteins. These molecules undergo phosphorylation and form a complex with PI3K. PI3K phosphorylates PIP2, which results in PIP3 formation and activation of PDK1. AKT gets phosphorylated and activated by PDK1. The latter is responsible for GLUT4 translocation to cellular membrane and glucose inflow. CDK4 was shown to activate the insulin-signaling pathway through phosphorylation of IRS2 at Ser388 upon insulin stimulation, thus maintaining insulin action in adipocytes. Although E2F1 and p53 are thought to interact with the insulin-signaling pathway to modulate AT insulin sensitivity, yet the underlying mechanisms are still unclear.

**Figure 4 biomolecules-10-01350-f004:**
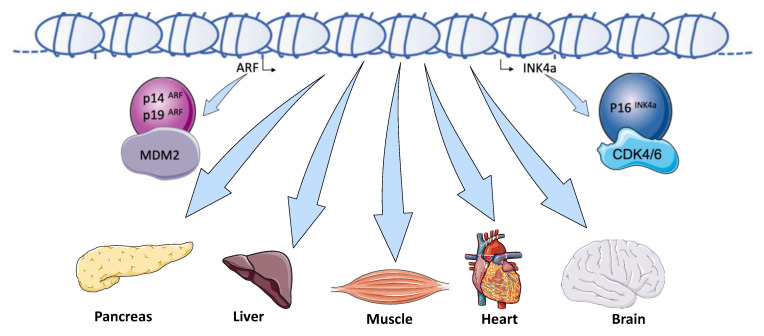
The *INK4a/ARF* locus: an emerging key actor in metabolic functions. In addition to AT, the *INK4A/ARF* (*CDKN2A*) locus is thought to play a variety of role in metabolic functions under normal and physiopathological conditions in other organs (pancreas, liver, muscle, heart, and brain). This locus affects glucose homeostasis, β-cell functions and mass, hepatic gluconeogenesis and lipid storage as well as cardiovascular functions. It also regulates the circadian rhythm, neurogenesis, and axonal regeneration.

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
