# Peer review of "Emerging Roles for the INK4a/ARF (CDKN2A) Locus in Adipose Tissue: Implications for Obesity and Type 2 Diabetes"

_biomolecules, 2020, doi:10.3390/biom10091350_

Round 1

Reviewer 1 Report

This is a fascinating topic. I often find in the literature that cell cycle molecules and their regulators are only discussed within a particular context, such as senescence only, or differentiation only. Seeing alternate roles for these molecules within a single tissue type/s discussed within one publication is certainly needed and appreciated. However this is a very technical topic and spending more time and perhaps adding an additional section introducing the reader into how these molecules work (and re-making figure 1) could be helpful. This review article is currently very difficult to read.

Figure 2 is your graphical abstract, but this is not even mentioned or described in the introduction. Use Figure 2 (or adapt it in such a fashion) so that a reader can follow the structure of the text using this figure as a map. Your sub-section titles should match this abstract (and do for the most part) and you should introduce the material in the Introduction section in this order. Finally, the structure of your Discussion section should follow this same order using the same themes and terminology to summarize what you have described to us in the body of the review. For Figure 2, numbering the 4 arms (senescence, inflammation, browning…) and matching those numbers to the numbers used in your sub-titles of your sections will help immensely to guide the reader. Even directly stating “The topics of this review will cover 1….2…3…and can be found in the graphical abstract in figure 2” will work. The other figures do not help very much either. A summary figure for each of the sub-sections would improve readability.

Because of the lack of clarity, lack of a distinct structure and a lack of appropriate citations, primarily in the introduction and in Section 5 (role in adipose inflammation), I suggest a partial rewrite. In general, the content is phenomenal but presented with too much concision for the breadth of information covered. Because of this, the impact this article will have for researchers who are not already familiar with these works is limited in its current form.

Introduction:

Good introduction with focused information to guide reader into the review well, however it appears to be missing many references pertinent to the claims made, and seems to gloss over some very technical information without enough explanation for someone who is not specifically an expert in cell cycle regulation. Line 59-79 in the introduction contains an overwhelming amount of information in a dense space. Perhaps splitting up this paragraph to contain genetic information and risk associations in one paragraph, and basic functions of cell cycle inhibition would have set me up better to receive the subsequent sections. For example, Figure 2 is fascinating and is your graphical abstract. This should be referenced at the end of the introduction and listing each of the 4 arms with a sentence for itself would help: such as “Cell cycle inhibition is important for balancing adipogenesis and senescence, but too much leads to…. “

-Line 29: Reference missing from definition of Obesity. This reviewer suggests including this one, or similar:

Lam YY, Ravussin E. Indirect calorimetry: an indispensable tool to understand and pre- dict obesity. Eur J Clin Nutr 2017;71:318-322.

-Line 29: Attributing Obesity solely to energy balance, although true, is an oversimplification that detracts overall from the introduction. Provide at least a few phrases describing other contributors to obesity such as genetics, obesogenic environmental influences (xenobiotics and ultraprocessed foods) that increase inflammation. This is very important for properly introducing your inflammation sub-section

-Line 31: Needs at least 1 citation demonstrating inflammatory profile associated with adipose tissue gain.

-Line 35: Cite types of adipose

-Many more citations needed in the introduction

Section 2

This section is very important, and still part of the introduction it seems. Please rework the information from this section into Figure 1. Then switch figure order to make figure 2 (graphical abstract) figure 1, and this one the new figure 2.

Section 3

This subsection is of most interest to me, however could be enhanced and made to be more complete. Line 170 needs a citation. Line 171-172 is incomplete, and when finished will need a citation. More information on connecting aging to obesity as far as senescence, and the impact of that senescent phenotype needed. Line 176 – What happens in AT that causes failure to expand? Any information on whether the cell cycle changes are causative or resultant? Line 176 would be a good place to allude to one of your next sections (inflammation). Paragraph 2 (line 177-188) is very cool!

Section 4

I already forgot where CDK4 comes in. This is a section that would really benefit from a diagram showing how these cell cycle regulators signal to enhance the insulin signaling pathway. You mention CDK4 acts in a cell-cycle-independent manner, but one more sentence reminding the reader of its “canonical” role in cell cycle before introducing this new role, would be helpful. Line 198: ‘exhibit’ misspelled.

Section5

Bringing aging back into the discussion to compare phenotypes of low-grade inflammation to obesity could be a way to make this section thorough. This section is poorly cited. Start with works on meta-inflammation and fibrosis with obesity (Daniel Lark or David Wasserman), and inflammation and macrophages with aging (Micah Drummond) or obesity (Ryan O’Connell).

Discussion:

I appreciate the strategy of summarizing and commenting on how to target these pathways for each specific function, as well as the challenges. However, at the start of the discussion section, please restate and summarize your graphical abstract (figure 2).

Reviewer 2 Report

A generally well written and informative review. My main concern is in regards to Figure 2, the authors reference Figure 2 in such a way that it suggest that Figure 2 should show how p53 is involved in adipogenic differentiation, how insulin acts on adipocytes, how adipocytes are sensitized to inflammation, etc. but the figure doesn't specifically show any of this. I would suggest changing where and how the figure is referred to, or changing the figure itself so that it is more specifically shows these. 

There are also a few typographical issues, see lines 172, 222-223, 229-230. 

Also in figure 2 the one adipocyte cell in the bottom left of the figure if very low resolution for some reason. 

Reviewer 3 Report

Dear authors,

The present review is very interesting and relevant in the field. However, I would like to make some suggestions in order to improve the manuscript. 

After reading the manuscript I think that the title is not appropriate for the paper content. Since the paper is concerned on adipose tissue, it should be mentioned on the title. Additionally, the control of energy homeostasis must include some metabolic markers, mainly the activation of AMPK. Is there some evidence on the association of INK4a/ARF locus on AMPK? 

On the introduction section, it should be interesting to include some information regarding the ectopic lipid accumulation that happens in the tissues instead of lipid accumulation in the AT depots.

Line 47, the authors need to describe the conditions in which the brown AT is present and is important to produce heat.

Line 61, it should be used T2D since the abbreviation was previously inserted.

Line 63, is important to include some knowledge about epigenetic to improve the reading about DNA methylation and be more clear to readers.

Line 104, please replace the sentence for "conversion of the stem cells into preadipocytes".

Lines 119-121, the sentence is not clear. Please re-write.

Line 172, The sentence is incomplete.

Line 198, please replace exhibite for exhibit.

After this revisions, I consider that the review is suitable for publication on Biomolecules.

Round 2

Reviewer 1 Report

This is a great article of a very difficult topic.  This reviewer appreciates the authors' efforts to (greatly) improve the readability.

One thing I’m still confused about is CDK4 (pRB) vs MDM2 (p53) regulation. Is it as simple as the pRB arm promoting differentiation of adipocytes, whereas activation of the p53 arm results in early senescence? I struggle with the regulation/fate decision between senescence and differentiation specifically. I wonder if it’s possible that maybe senescence is a form of premature differentiation due to epigenetic changes preventing suppression of differentiation transcriptional program (inability to maintain quiesence)? This can be due to inflammation and other cues eventually causing histone reductions within stem cells, for example. If you think this question makes sense or could be relevant, please add 1-2 sentences addressing it in section 2 or the discussion to make this portion of the review more satisfying.

Minor comments:

Line 73: DT2 should be changed to T2D for consistency

Line 235-236: Fix Grammar: "Although metabolic insults directly up-regulates….yet the underlying"

Line 295-296: “In short” not needed for figure legend description.
